# Electroencephalography-Based Effects of Acute Alcohol Intake on the Pain Matrix

**DOI:** 10.3390/brainsci13121659

**Published:** 2023-11-30

**Authors:** Elias Dreismickenbecker, Sebastian Zinn, Mara Romero-Richter, Madeline Kohlhaas, Lukas R. Fricker, Silvana Petzel-Witt, Carmen Walter, Matthias Kreuzer, Stefan W. Toennes, Malte Anders

**Affiliations:** 1Center for Pediatric and Adolescent Medicine, Department of Pediatric Hematology/Oncology, University Medical Center Mainz, 55131 Mainz, Germany; 2Clinical Development and Human Pain Models, Fraunhofer Institute for Translational Medicine and Pharmacology ITMP, 60596 Frankfurt, Germany; mara.romero.richter@outlook.de (M.R.-R.); lukas.r.fricker@gmail.com (L.R.F.); carmen.walter@itmp.fraunhofer.de (C.W.); malteanders@gmail.com (M.A.); 3Department of Anesthesiology, Intensive Care Medicine and Pain Therapy, Goethe University Frankfurt, University Hospital, Theodor-Stern-Kai 7, 60590 Frankfurt, Germany; zinn@med.uni-frankfurt.de (S.Z.); madeline.kohlhaas@t-online.de (M.K.); 4Institute of Legal Medicine, University Hospital, Goethe University, 60590 Frankfurt, Germany; witt@med.uni-frankfurt.de (S.P.-W.); toennes@em.uni-frankfurt.de (S.W.T.); 5Department of Anesthesiology and Intensive Care, School of Medicine and Health, Technical University of Munich, 81675 Munich, Germany; m.kreuzer@tum.de

**Keywords:** alcohol, pain, nociception, noxious stimulation, ethanol, EEG, pain matrix

## Abstract

The effects of acute and chronic intakes of high doses of alcohol on pain perception are well known, ranging from short-term analgesic effects to long-term sensitization and polyneuropathies. The short-term analgesic effects of ethanol consumption on subjective pain perception have been well studied in the literature. Recent advances in neuroimaging allow for an insight into pain-related structures in the brain, fostering the mechanistic understanding of the processing of nociceptive input and pain. We aimed to utilize EEG, combined with standardized noxious mechanical/thermal stimulation and subjective pain testing, to research the effects of acute alcohol intake on nociceptive processing and pain perception. We recruited 12 healthy subjects in an unblinded cross-over study design and aimed at achieving a blood alcohol level of 0.1%. Our data revealed a significant reduction in subjective pain ratings to noxious thermal and mechanical stimuli after alcohol ingestion. Our EEG data revealed suppressing effects on the cortical structures responsible for processing pain, the “pain matrix”. We conclude that in addition to its analgesic effects, as expressed by the reduction in subjective pain, alcohol has a further impact on the “pain matrix” and directly affects the salience to a nociceptive stimulus.

## 1. Introduction

The analgesic properties of alcohol, i.e., ethyl alcohol, are a long-standing belief and have been studied in the past [1,2,3]. However, actual study data do not provide conclusive evidence of a causal relationship between alcohol consumption and analgesic effects. Experimental studies have yielded inconsistent results, partly due to the presence of subjective assessment bias. Though some studies have revealed evidence of alcohol-induced analgesia, the effect size varies greatly and the results are mixed [3,4]. Furthermore, the comparability of studies is limited by methodological restrictions such as the lack of a placebo control group, no blinding, or small numbers of subjects [4].

Nevertheless, several possible mechanisms have been proposed. Most of the literature points towards modifications in neurotransmitters and neuropeptides such as GABA, glutamate, and endogenous opioids [5]. However, certain authors advocate for indirect pain mechanisms, like anxiety relief or affective components [6]. Also, it is conceivable that there are shared neural circuits between chronic pain and alcohol dependence and that pain conditions influence alcohol consumption by affecting the reward pathways that regulate consumption [3]. Unsurprisingly, alcohol is used for the self-management of pain [4,7], although it is an unsuitable analgesic, having adverse effects such as addiction and illness in the long term [5]. This is reinforced by the fact that consuming doses of alcohol that are characterized as “safe drinking” [8] may not be suited to relieve pain; the literature points towards a minimum blood alcohol level of 0.08% for achieving an analgesic effect [3]; however, this level has been shown to have long-term damaging effects on the body if consumed chronically [9]. In addition, in the long-term, the chronic use and abuse of alcohol increases the sensitivity to nociception and leads to hyperalgesia [10], which further limits its use as an analgesic.

The use of experimental pain paradigms can help determine causality by examining the effect of measured doses of alcohol on quantifiable pain indices in response to noxious stimuli [3]. Several studies have investigated the effect of alcohol on subjective pain perception and the endogenous pain modulation mechanisms [1,2,3,4,11]. Given the inherent subjectivity of self-reported data, electroencephalography (EEG) has emerged as a more objective method to address these limitations [12]. Since the conscious perception of pain originates from the nociception of higher brain centers, the neural process of encoding noxious stimuli can occur without an individual labeling it as “pain” [13]. Conversely, pain does not always result from the processing of a noxious stimulus [14,15]. While the activation of nociceptors plays a significant role, pain always requires subjectivity, which requires consciousness and the ability to evaluate a stimulus or situation [16]. Pain is subject to modulation across the central nervous system and is influenced by non-nociceptive aspects of sensory input, along with cognitive and affective/emotional factors.

Therefore, incorporating EEG could greatly enhance our understanding of subjective ratings of noxious stimuli by providing a means to quantify pain-related brain responses as a reflection of the processing of such stimuli. Research has shown that increased subjective pain sensations correspond to a decrease in specific frequency bands within EEG recordings [17]. While self-reported subjective pain testing is still viewed as the gold standard in pain assessment [18], it may not unmask the underlying mechanisms and why alcohol is sometimes preferred over, or in combination with, other drugs when self-managing pain. Also, analyses of subjective pain ratings do not allow for an interpretation of whether observed effects are possibly caused directly by alcohol consumption or indirectly by affective components of alcohol consumption such as anxiety relief.

Human brain imaging using, for example, electroencephalogram (EEG) or functional magnetic resonance imaging (fMRI) delivers promising tools to research the neuronal mechanisms of pain. However, the validity of those tools as biomarkers for pain is still under discussion [19,20,21,22]. In any case, while EEG does not serve as an objective biomarker for pain intensity [20,21,23], it can help uncover processes in the cortical regions responsible for processing nociceptive information [21,24]. Hence, while still influenced by other factors such as attention towards the stimulus (stimulus salience), EEG fosters our understanding with its indirect readout of the processing of nociceptive stimuli [20]. Furthermore, a recent study researching the acute intake of alcohol using fMRI points towards an effect of acute alcohol consumption on the reward-, emotion-, and motivation-regulating structures in the brain, proposing further mechanistic effects at the neuronal level [25].

With our study we aim to provide an insight into the acute effects of alcohol intake on the cortical structures responsible for processing noxious information, the “pain matrix” [24], using EEG and standardized nociceptive stimulation. For that purpose, we analyzed the event-related spectral perturbation (ERSP) patterns in the N2P2 EEG response at the Cz electrode following noxious stimulation. Our aim is to leverage these patterns to draw conclusions regarding the pain matrix; a network of brain regions that is activated in response to nociceptive stimuli and contributes to pain perception. It has been utilized in functional imaging research to identify associations between brain activity and pain processing [26]. While the precise mechanisms through which alcohol influences pain perception are still unclear, analyzing the pain matrix could provide valuable insights into this phenomenon [4].

We investigate the influence of alcohol on evoked potentials of the sensory cortex as an interface between nociception and pain processing. We recruited healthy adults in an unblinded cross-over design study, where one group consumed alcohol until they reached a calculated blood alcohol level of 0.1%, and the other consumed a placebo drink. In addition, we tested the subjective pain ratings to our standardized stimuli to verify if we had achieved a sufficient analgesic effect. In light of recent research indicating a correlation between individuals’ subjective pain sensation and the extent of central activation, our hypothesis postulates that the group exposed to alcohol will exhibit reduced activation levels within the event-related spectral perturbation graphs attributable to the analgesic properties of alcohol. If alcohol shows suppressive effects on the pain matrix, it may help to understand why alcohol is often used as an analgesic for pain. This could highlight the need for promoting alternative treatments for patients that carry fewer negative health outcomes.

## 2. Materials and Methods

### 2.1. Study Protocol and Participants

We recruited healthy male and female volunteers with a minimum age of 18 years and a maximum age of 35 years. Due to our explorative study approach and the lack of preliminary research, we did not perform an a priori sample size calculation [27].

The participants had to express their willingness to participate in the study via a written consent form. The inclusion criterion was experience of consuming alcohol; this was defined as having consumed at least one standard drink per week on average for the past three months, but not more than 7 drinks per week (female participants) or 14 drinks per week (male participants). The exclusion criteria included alcohol or drug abuse in the past or present, illnesses of the central nervous system, metabolic diseases, an allergic disposition to alcohol, a known polymorphism in the alcohol dehydrogenase enzyme, or the current or frequent intake of painkillers or psychotropic medication. A 10-panel drug test was carried out before each study day: participants who had a positive result for any of the tested substances were excluded from the study. Female participants also had to undergo urine-based pregnancy testing (WiduMed hCG Schwangerschafts-Teststreifen, Rödinghausen, Germany).

Each participant underwent the study flow two times on two different days in a pseudo-randomized fashion. The study sessions took part in the morning of the respective day; the subjects were asked to refrain from eating breakfast and drinking anything besides water as they received a standardized breakfast on-site (5 slices of rusk and 250 g of full-fat yogurt). Following breakfast, the baseline recordings took place. Afterwards, the participants consumed either alcoholic beverages with an alcohol content of 10% (*v*/*v*), comprising of pharmaceutical grade ethyl alcohol, carbonated water, lime juice, and brown sugar (alcohol day) or an equal volume of a placebo drink, where the pharmaceutical grade ethyl alcohol was not added to the drink (placebo day). The participants were unblinded to which drink they were consuming as the effects and taste of the drinks would make a blinding process impossible. The amount of alcohol that was to be consumed on the alcohol day was calculated according to the Widmark formula [28] to reach the target blood alcohol level of 0.1% [29]. For this calculation, a Widmark factor (representing the volume of distribution) of 0.6 was used for the female and 0.7 for the male subjects. Furthermore, it was assumed that only 70% of the consumed alcohol would be adsorbed (representing the first-pass effect). The drinks were consumed over a period of 45 min in three tranches, where one third of the overall volume had to be finished in each respective 15 min. This was to achieve similar drinking speeds and reduce the typical inter-individual variation in absorption. After the participants had finished the drinks, a 20 min period passed before further experimental procedures to allow for alcohol absorption to proceed. Following this, Block 1 of the testing began with a test of the breath alcohol level using an Alcotest 3000 device (Dräger Safety AG & Co., KGaA, Lübeck, Germany). Subsequently, the next block of noxious stimulation was administered, and blood was drawn immediately after the stimulation block was finished to determine the blood alcohol concentration (BAC). We then waited 60 min between Block 1 and Block 2, and Block 2 and Block 3, respectively, and again started Block 2/3 by analyzing the breath alcohol level.

Blood sampling was performed directly after each stimulation block to capture the minimum blood alcohol level at which the stimulation was performed. Blood alcohol concentrations (BAC) were determined by headspace gas chromatographic analysis of 0.25 mL serum following separation by centrifugation using a validated forensic routine method [30]. Alcohol serum concentrations were divided by 12.36 (the distribution ratio between the serum and whole blood) to be expressed as a % in whole blood, which is a commonly used unit for blood alcohol concentrations. The reliability of this method has been regularly confirmed by proficiency testing (Arvecon, Walldorf, Germany).

### 2.2. Standardized Noxious Stimulation

During each of the four blocks, the participants underwent standardized noxious stimulation consisting of noxious heat using a contact heat stimulator (MEDOC PATHWAY Pain and Sensory Evaluation System, Medoc Limited, Ramat Yishai, Israel) with 54 °C peak temperature and noxious mechanical stimulation using a 512 mN pinprick (MRC Systems GmbH, Heidelberg, Germany). We aimed at recording contact heat-evoked potentials (CHEPS) and pinprick-evoked potentials (PEP) by stimulating the dorsum of the dominant hand in each case. Both devices were synchronized with the EEG device via a +5 V TTL pulse, with the pinprick achieving this via an optical trigger, an LM393 (Texas Instruments, Dallas, TX, USA), and an ATmega32U4 (Microchip, Chandler, AZ, USA), as described in the literature [31]. The levels of stimulation energy consisted of a peak temperature of 54 °C during noxious heat stimulation and a force of 512 mN during mechanical pinprick stimulation; these were chosen as they are above the respective mechanical and heat pain thresholds in the QST reference data for 15–35-year-old participants [32]. In this way, we ensured that our stimuli would be described as painful by a healthy participant in our age group of 18–35 years. Each stimulus was applied 12 times in a randomized pattern over the stimulation area across the whole dorsum of the hand to avoid habituation. We set the inter-stimulus interval to 8–12 s for the noxious contact heat stimulation and to 3–5 s for the noxious mechanical stimulation. We asked the participants to rate each stimulus on a verbal analog scale (VAS) from 0 to 100, approximately 2 s after each stimulus, where 0 indicated no painful sensation, and 100 indicated the maximum bearable pain.

### 2.3. EEG Recording and Pre-Processing

We recorded EEG from 64 active electrodes (g.Tec g.SCARABEO and g.Tec g.HIamp, Guger Technologies, Schiedlberg, Austria) arranged according to the 10–20 system. The electrodes were initially referenced to Afz and distributed equally across the scalp. Following the recordings, we used EEGLAB [33] to downsample the EEG to 256 Hz for data reduction and applied a zero-phase bandpass filter between 1 Hz and 100 Hz. We reduced the 50 Hz line noise using the EEGLAB plug-in CleanLine and rejected artifacts using artifact subspace reconstruction (ASR) with a tolerance parameter of 20 [34]. We visually rejected corrupted channels (i.e., due to defective electrodes) and interpolated the removed channels by using spherical spline interpolation [35]. The datasets were then re-referenced to average reference. We then epoched the data from −1 s to +2 s around the onset of each stimulus and subsequently calculated the event-related spectral perturbation (ERSP) and the inter-trial coherence (ITC) using EEGLAB’s newtimef-function with a divisive baseline from −1 s to 0 s, a resolution in time of 400 points from −1 s to +2 s, and a frequency resolution of 200 points between the frequencies of 3 Hz and 100 Hz [36,37,38]. We ran the wavelet transform portion of the newtimef-function with three cycles at the lowest frequency of 3 Hz and 20 cycles at the highest frequency of 100 Hz and analyzed the data at the electrode location Cz [31,36,39,40].

### 2.4. Statistics

Due to our small sample size of 12 participants, we used non-parametrical approaches to describe the statistical differences in our data [41]. To compare the VAS values, we applied the Friedman test for three or more groups of dependent data. For post hoc testing, we used the MATLAB (Mathworks Version R2021a. 9.10.0.1602886, Natick, MA, USA) function multcompare. To test for statistical differences in the EEG, we calculated the area under the receiver operating characteristics (AUROC) with the 1000-fold bootstrapped 95% confidence intervals using the MES toolbox for MATLAB [42]. We considered differences to be significant if the 95% AUROC confidence interval did not include 0.5 [42,43]. The AUROC effect size reveals a perfect separation between groups or conditions if the value is either 1 or 0, while a value of 0.5 indicates no separation between the groups [43]. Furthermore, by either being 1 or 0, the AUROC indicates the direction of the effect. In addition, we rated our AUROC effect size according to a traditional point system; the range of 1–0.9/0–0.1 is rated as an excellent effect/an excellent separation, between 0.9 and 0.8/0.1 and 0.2 it is rated as a good effect, while a range of between 0.8 and 0.7/0.2 and 0.3 is rated as a fair effect. Everything below this is rated as clinically irrelevant as it is characterized as being a poor separation [44]. We used statistical approaches that test dependent data as we compared the same subjects during different study days (groups) and conditions. Hence, we compared the relative AUROC changes between the two conditions and compared these to a fixed value of 1 using the auroc function of the MES toolbox [42]. For reasons of comprehensibility when extracting the ERSP values, we have only presented the maximum ERSP values and their respective 25% and 75% quartiles, as well as the time and frequency at which they occurred. This approach is not dependent on the chosen window size when extracting the ERSP data.

To account for multiple comparisons, instead of a common approach of an alpha level adjustment, we have applied a cluster-based approach as it has been used in the literature both for 2-dimensional [45,46] and 3-dimensional [47,48] EEG data. We have only reported results as being significant if they appeared in clusters of at least 3 × 3 pixels in size [36].

## 3. Results

### 3.1. Participants and Alcohol Levels

We recruited twelve participants (seven male, five female) between the ages of 22 and 29 years, with a median age of 25.6 years. The median blood and breath alcohol levels at the respective recording points are outlined in Table 1. In ten out of twelve participants, the blood alcohol levels declined between Block 1 and Block 2, while they still increased in two participants. After Block 2, the blood alcohol levels steadily declined until Block 3 in all participants.

### 3.2. Subjective Pain Ratings

In Figure 1, we show the median VAS values obtained for all 12 participants as a median over all 12 trials of noxious stimulation. We compared these values using Friedman’s test. For comprehensibility during post hoc testing, we only show the statistical comparisons and their respective *p* values between Block 1 and 3 versus the baseline value.

### 3.3. Event-Related EEG Data as Spectral Perturbation

We show the event-related spectral perturbation (ERSP) as a response to our standardized noxious stimulation with noxious contact heat in Figure 2 and to our noxious mechanical stimulation with a pinprick in Figure 3. For comprehensibility, we only show the data and statistics for comparing the Baseline vs. Block 1 as we achieved the highest median alcohol levels in that block.

On average, our participants elicited a response, i.e., an activation or a deactivation, which exceeded 2 dB or −2 dB at different points in time and frequencies for the different stimulation techniques. In general, for CHEPS, we observed most of the responses in the lower frequency regions, approximately from 1 to 10 Hz between 300 and 500 ms following stimulus onset (black arrow in Figure 2). For PEP, this lower frequency response had an earlier onset, approximately 50–150 ms following stimulus onset (black arrow in Figure 3). In the alcohol group during the baseline recordings, we observed an additional response in the higher frequency regions from approximately 18 to 24 Hz between 1000 and 1200 ms following stimulus onset, indicated by a white arrow in Figure 3. We show the maximum ERSP values with the 25% and 75% quartiles and the corresponding times and frequencies for the low frequency responses in Table 2.

For CHEPS, our statistics revealed spots of significance between the Baseline and Block 1 conditions in the placebo group (P0 and P1); the maximum AUROC effect size with the confidence intervals in square brackets for this comparison was 0.83 [0.58; 1] at 7.9 Hz and 519 ms, indicating a good effect as per our traditional points system. In total, 215 pixels were significantly different between the conditions. Subsequently, the statistics between the Baseline and Block 1 in the alcohol group (A0 and A1) also revealed spots of significance, with a minimum AUROC value of 0.17 [0; 0.42] at 4.9 Hz and 367 ms, also indicating a good effect; the pixel count for this statistical difference was 60. As for the inter-group comparison P0 vs. A0, the spot of significance (41 pixels) had a maximum AUROC value of 0.75 [0.51; 0.99] at 8.4 Hz and 437 ms, indicating a fair effect. Subsequently, for P1 vs. A1, the spot of significance (602 pixels) had a minimum AUROC value of 0.08 [0; 0.25] at 10.3 Hz and 469 ms, indicating an excellent separation between the groups.

For PEP, our statistics revealed a significant difference (85 pixels) in the lower frequency regions between conditions P0 and P1, with a maximum AUROC value of 0.83 [0.58; 1] at 5.9 Hz and 27 ms, indicating a good effect. Subsequently, the inter-group comparison between P1 and A1 in the lower frequency regions revealed a significant difference (208 pixels) with a minimum AUROC value of 0 [0; 0] at 4.7 Hz and 105 ms, indicating a perfect separation.

## 4. Discussion

In this study, we demonstrate a significant reduction in subjective pain ratings following standardized noxious stimulation in participants who consumed considerable amounts of alcohol. This acute analgesic effect of alcohol has been shown in the literature [3,4], while the exact mechanism of action is still being debated [5,11,49]. Our subjective pain testing was extended with the objective pain signatures extracted from the EEG following standardized noxious stimulation. We aimed to explore the effects of acute alcohol intake on the pain matrix, i.e., the cortical network (including the somatosensory, insular, frontal, parietal, and cingulate structures) that is responsible for the processing of nociceptive information in the brain [24].

The increase in power as a response to the stimulus in the placebo group between the Baseline and Block 1 (P0 vs. P1 for both methods of standardized noxious stimulation), where no intervention took place, may be explained by other salient sensory events since the recent literature suggests that the pain matrix might not be solely specific to pain [50]. Traditionally, the “pain matrix” refers to the network that presents the intensity and unpleasantness of perception caused by a nociceptive stimulus. However, it is important to note that nociception and pain are not interchangeable as pain can arise in the absence of nociceptive input, and not every nociceptive input results in subjective pain perception. Therefore, possible explanations for our findings might be either intra-subject variability [20,21,51], adaptive processes of the peripheral nervous system (sensitization) to the stimulus [52,53], or some kind of learning effect that altered the salience, defined as the ability of the stimulus to capture the attention of the participant, to the stimulus [20]. The descriptive observation of an initial increase in subjective VAS ratings in Block 1 within the placebo group where no intervention took place, followed by a potential habituation in subsequent blocks, could also support the hypothesis of an influence of affective or emotional effects (e.g., stimulus salience), which might be further compounded by the non-blinded study design.

An increase in subjective pain sensation has been associated with a decrease in power across various frequency bands, especially alpha2, beta1, and beta2 [17]. Additionally, a power decrease appears to indicate an altered pain modulation. The power increase in the placebo group (P0 to P1) as a response to the noxious stimulus could therefore indicate a decreased subjective pain sensation due to endogenous pain modulation [17]. However, this cannot be confirmed looking at the subjective VAS ratings, potentially indicating the benefit of EEG to complement subjective reports.

Interestingly, we observed no increase in ERSP power in the alcohol group from the Baseline to Block 1 (A0 vs. A1); the power decreased for both types of noxious stimuli, but only significantly during CHEPS. Although the difference is noticeable in the average EEG signatures following pinprick stimulation in Figure 2, it is not significantly different. A comparable trend emerges when contrasting the ERSPs of the alcohol and placebo groups in block 1 (P1 vs. A1). Notably, the placebo group demonstrates significantly higher power in the ERSPs within both CHEPS and PEP. This suppression of the pain matrix in the alcohol group in combination with the significantly reduced VAS ratings may signify the direct suppressing effects of alcohol on pain and pain modulation [26].

However, as previously stated, a reduction in central activation following noxious stimuli, like observed in the alcohol group, can also be associated with an augmented subjective perception of pain [17], but the subjective VAS ratings in the alcohol group are not consistent with this observation as they show significantly decreased values after alcohol consumption that do not support an increased pain sensation. This could imply that while alcohol indirectly suppresses pain perception through affective factors, leading to lower VAS scores, it also leads to a central deactivation that was associated with higher subjective pain experience in other subjects. Given the limited spatial resolution of the EEG and our limited number of subjects, further interpretation is hindered and would be speculative at this moment. Therefore, a beneficial next step would involve employing a procedure capable of localizing the observed central phenomena.

The fact that we could only find significant differences in the comparison of the CHEPS might be explained by the small sample size and our conservative statistical approach. Furthermore, the available literature suggests that the analgesic effects of alcohol start at a blood alcohol concentration of 0.08% [3], although two of our participants did not exceed this level during Block 1 testing. Thus, we propose that if a stimulus sensitization has occurred, it is counter-acted by alcohol and that the stimulus response in the cortical regions representing the “pain matrix” [24] is further suppressed due to additional effects beyond a simple analgesia, such as a decreased stimulus salience [20]. This is further underlined when comparing the placebo group to the alcohol group during Block 1 (P1 vs. A1); both statistics reveal big spots of significant differences with excellent AUROC effect sizes close to zero (CHEPS) or even a perfect separation in all 12 participants, as per our AUROC model during the pinprick stimulation. The processing of our standardized noxious stimuli, when applied a second time after a Baseline recording, is thus significantly suppressed by the intake of alcohol when comparing the alcohol group with the placebo group. In addition to those cortical effects, the commonly known analgesic effect is substantiated by the significant reduction in the subjective pain ratings in our data. As the composition of the EEG response is a combination of the painfulness of the stimulus and the stimulus salience, this can either be a result of the reduction in pain, as expressed by the subjective pain ratings, or be due to a reduction in the salience to the stimulus [20,23,24], as expressed by our EEG data. Although it is a possibility, we cannot conclude a sensitization to our nociceptive stimuli from our data between the Baseline and Block 1. Nevertheless, further research could test if this would also be counter-acted by alcohol, as our data do allow for this hypothesis to be postulated.

## 5. Conclusions

We conclude that the acute intake of alcohol also has a suppressing effect on the activation of pain-related brain structures, the so-called pain matrix, and systems that are described as the “salience detection system for the body” [24]. A decreased salience may be understood as a decrease in the intensity and unpleasantness of the perception of our standardized nociceptive stimuli [24]. Subsequently, it is important to understand that the pain intensity is not always associated with the strength of the response in the “pain matrix” [20,24]. In the case of acute alcohol consumption, however, both the pain levels as per subjective pain ratings and the response of the “pain matrix” are suppressed. Hence, it is likely that alcohol, in addition to its analgesic properties, has more subtle effects on the processing of pain in the cortical regions that may not be captured with standardized pain testing methods alone but that can be unmasked with neuroimaging tools such as EEG. Consequently, there is a compelling need to sensitize vulnerable patients to alternative, less-harmful treatment methods. Exploring and promoting these alternatives could prove valuable in addressing chronic pain without the potential drawbacks associated with alcohol consumption.

## 6. Limitations

Our results are limited by the small sample size, unblinded study design, and the intra-subject fluctuations concerning the pharmacokinetics of orally administered alcohol. In addition, due to the small sample size, we adhered to a rather conservative non-parametric statistical approach, which may not have unmasked the full extent of the group differences.

Additionally, precise localization of central effects was unattainable due to the EEG’s constrained spatial resolution, preventing the assignment of phenomena to specific functional areas. Furthermore, we did not collect any further data on the regularity of alcohol consumption by our test subjects, which could also have influenced our results as a different effect on the amount of alcohol can be expected between individuals.

Also, our measurements were conducted on pain-free subjects. Subsequently, exploring alterations in individuals experiencing pain would be a compelling next phase of investigation since chronic pain has an impact on central activation and deactivation, as well as the pain matrix itself.

## Figures and Tables

**Figure 1 brainsci-13-01659-f001:**
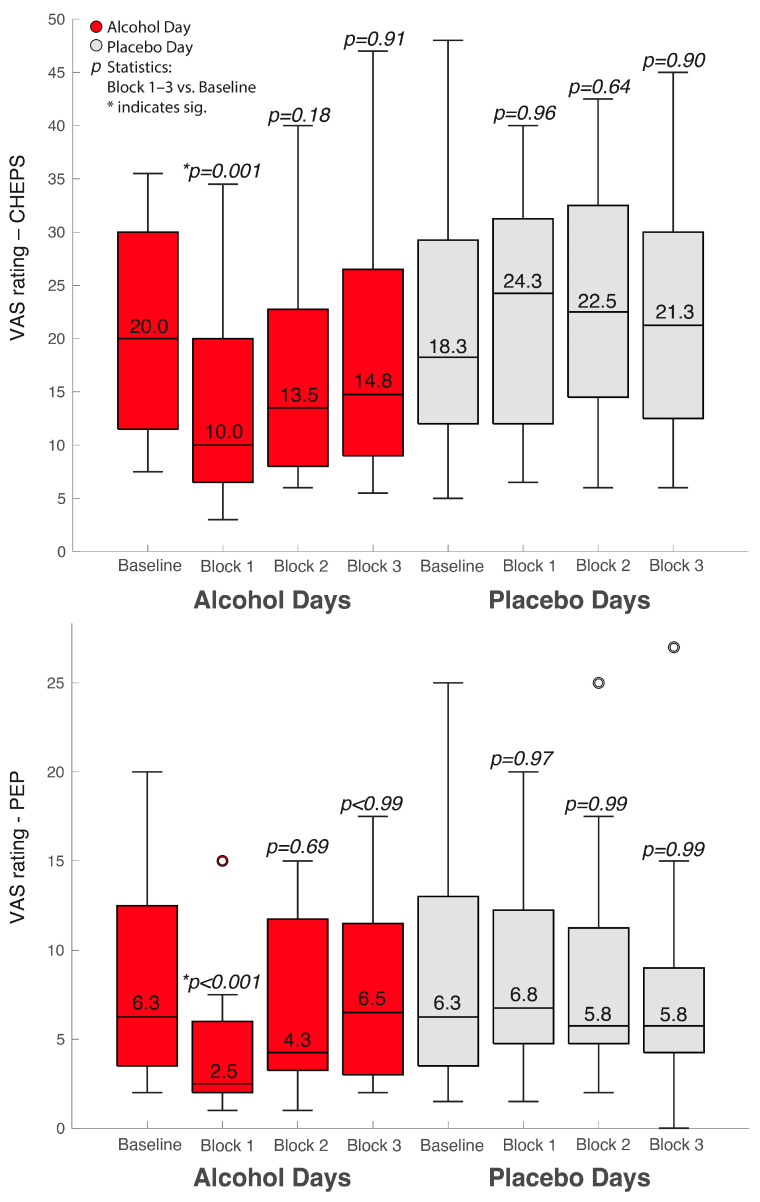
Median VAS values of our standardized noxious stimulations in the different groups and conditions. The upper boxplots represent the VAS values following CHEPS and the lower boxplot the VAS values following PEP.

**Figure 2 brainsci-13-01659-f002:**
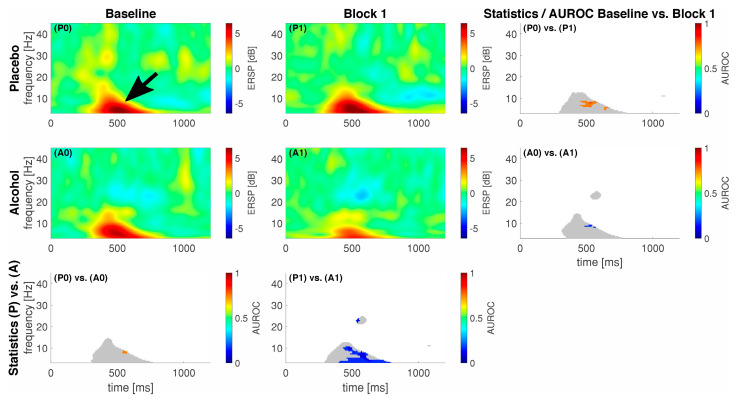
Eventrelated data following noxious contact heat stimulation (CHEPS) as spectral perturbation (ERSP). The gray shaded areas in the statistics panels indicate a response, i.e., either one of the areas in the respective ERSP patterns exceeded −2/2 dB; statistics were only calculated for those areas. Areas of interest for further analysis highlighted with a black arrow (early low-frequency response). A colored pixel in the statistics image indicates statistical significance; the color value according to the c-axis indicates the AUROC effect size. The EEG data are shown at electrode location Cz. A0 and P0 and A1 and P1 represent block 0 and 1 in the alcohol (A) and placebo (P) groups, respectively.

**Figure 3 brainsci-13-01659-f003:**
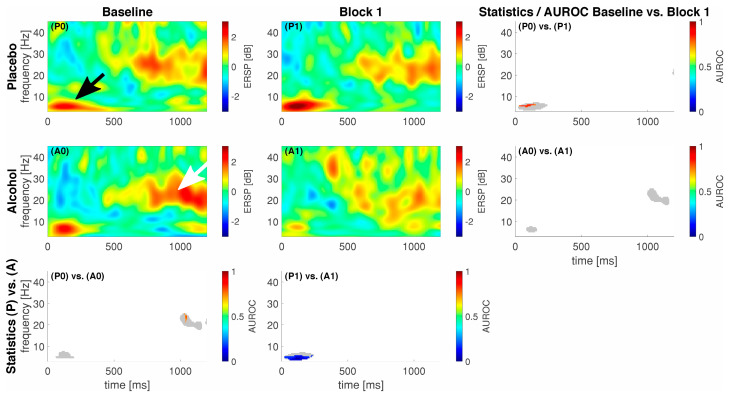
Event-related data following noxious pinprick stimulation (PEP) as a spectral perturbation (ERSP). The gray shaded areas in the statistics panels indicate a response, i.e., either one of the areas in the respective ERSP patterns exceeded −2/2 dB; statistics were only calculated for those areas. Areas of interest for further analysis highlighted with a black (early low-frequency response) and whihte arrow (late high-frequency response). A colored pixel in the statistics image indicates statistical significance; the color value according to the c-axis indicates the AUROC effect size. The EEG data are shown at electrode location Cz. A0 and P0 and A1 and P1 represent block 0 and 1 in the alcohol (A) and placebo (P) groups, respectively.

**Table 1 brainsci-13-01659-t001:** Median blood and breath alcohol levels and their respective 25% and 75% quartiles in brackets. The arrows indicate that the measurement types are listed in the table row and the measurement points are listed in the table column.

→ Measuring Type ↓ Measuring Points	Breath Alcohol Level (%)	Blood Alcohol Level (%)
Baseline: First measuring point	0 (0 to 0)	not tested
Block 1: Second measuring point	0.077 (0.073 to 0.090)	0.102 (0.088 to 0.115)
Block 2: Third measuring point	0.068 (0.065 to 0.074)	0.088 (0.083 to 0.094)
Block 3: Fourth measuring point	0.052 (0.048 to 0.056)	0.070 (0.065 to 0.075)

**Table 2 brainsci-13-01659-t002:** Maximum ERSP values, including the 25% and 75% quartiles in brackets, and the respective times and frequencies at which the maximum ERSP values occur.

	Placebo			Alcohol		
	Power (dB)	Frequency (Hz)	Time (ms)	Power (dB)	Frequency (Hz)	Time (ms)
PEP						
Baseline	2.94 (0.38 to 3.66)	5.4	129	2.42 (0.75 to 4.20)	5.9	129
Block 1	3.04 (2.15 to 4.20)	5.4	70	1.4 (−0.80 to 2.33)	4.9	90
CHEPS						
Baseline	7.99 (3.60 to 9.71)	4.9	469	7.82 (3.07 to 8.79)	4.5	496
Block 1	8.41 (3.10 to 10.23)	3.5	512	4.53 (2.89 to 5.68)	3.0	527

## Data Availability

The data that support the findings of this study are available from the correspondence author, upon reasonable request. The data are not publicly available due to participants’ privacy concerns. The data are stored in protected datasets in our institution.

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
