# Peer review of "Electroencephalography-Based Effects of Acute Alcohol Intake on the Pain Matrix"

_brainsci, 2023, doi:10.3390/brainsci13121659_

Round 1

Reviewer 1 Report

Comments and Suggestions for Authors

Please see in the attachment.

Comments on the Quality of English Language

Minor editing of the English language required

Reviewer 2 Report

Comments and Suggestions for Authors

This is an interesting paper that expands our knowledge on the effect induced by alcohol on pain and cortical nociceptive pathway. The Introduction section is well-written and clearly state the aim of study. The Material and Methods section described in detail ways and means employed. The Results section clearly reports what happened during the course of the experiments. Finally, the Discussion section effectively debate the results obtained and furnish interesting ideas for further experiments. In any case, a small revision of the article is necessary before publication and authors should consider the following suggestions.

·         In the version of the manuscript that I downloaded, the following sentence often appears: "Error! Reference source not found", such as on page 3, line 108, page 5, line 206, page 7, lines 236-238. Authors should check this message and possibly modify the text accordingly.

·         Page 3, line 132. Please change "blood alcohol concentric (BAC)" to "blood alcohol concentration (BAC)". The authors should also provide a description, even a brief one, of how this parameter was obtained.

Reviewer 3 Report

Comments and Suggestions for Authors

Manuscript title: The EEG-based effects of acute alcohol intake on the pain matrix

The study explores the impact of acute alcohol intake on the pain matrix using EEG measurements. Overall, I find the topic to be intriguing and relevant to the field; however, several points need to be addressed before the manuscript can be considered for publication.

Major comments:

1.     There are several pre-clinical studies indicating the development of alcohol-induced neuropathy. However, this study is considering the protective effects of alcohol. What is the author's take on this? Kindly go through the following literature and include a justification regarding the same in the manuscript

DOI: 10.1111/j.13652125.2011.04111.x

2.     There is an issue in the referencing throughout the manuscript. It is showing “Error! Reference source not found. – Kindly rectify the same.

3.     In table 2 the data related to the pinprick test is referred to as pin-prick where the same has been referred to as PEP in the figure 1 and 3. Kindly maintain the same terminology to avoid confusing the readers.

4.     The term “pain matrix” has been used throughout the manuscript, but there is no proper explanation of what constitutes the pain matrix and its components. It would be beneficial to include a brief definition to ensure that readers, including those not specialized in the field, can understand its significance.

5.     The authors must try to establish a direct correlation between the concept of the pain matrix and the objectives of the current study. Explain how the EEG-based measurements in the study relate to the components of the pain matrix. Having some clarity on this connection will help understand the rationale behind investigating acute alcohol intake in the context of the pain matrix.

6.     Discuss about the possible future investigations in this line of research.

7.     How was the particular dose of alcohol decided in the study? It is suggested to investigate the dose-response relationship by exploring different levels of alcohol intake. Examining varying alcohol doses may help delineate a more in depth understanding of how EEG patterns within the pain matrix respond to different levels of acute alcohol exposure.
